# Memory Retrieval in Transformers: Insights from the Encoding Specificity Principle

## Abstract

While explainable artificial intelligence (XAI) for large language models (LLMs) remains an evolving field with many unresolved questions, increasing regulatory pressures have spurred interest in its role in ensuring transparency, accountability, and privacy-preserving machine unlearning. Despite recent advances in XAI have provided some insights, the specific role of attention layers in transformer-based LLMs remains underexplored. This study investigates the memory mechanisms instantiated by attention layers, drawing on prior research in psychology and computational psycholinguistics that links Transformer attention to cue-based retrieval in human memory. In this view, queries encode the retrieval context, keys index candidate memory traces, attention weights quantify cue–trace similarity, and values carry the encoded content, jointly enabling the construction of a context representation that precedes and facilitates memory retrieval. Guided by the Encoding Specificity Principle, we hypothesize that the cues used in the initial stage of retrieval are instantiated as keywords. We provide converging evidence for this keywords-as-cues hypothesis. In addition, we isolate neurons within attention layers whose activations selectively encode and facilitate the retrieval of context-defining keywords. Consequently, these keywords can be extracted from identified neurons and further contribute to downstream applications such as unlearning.

## 1 Introduction

Transformer-based Large Language Models (LLMs) are often characterized as "black-box" systems due to the opacity of their internal processes. This lack of transparency raises concerns about safety, privacy, and accountability, thereby motivating the development of explainable artificial intelligence (XAI) (Zhao et al., 2024). While XAI aims to improve model interpretability, it does not directly address the challenges of data removal or user control over personal information. In parallel, the field of machine unlearning has gained traction as a complementary approach for mitigating privacy risks in LLMs (Jang et al., 2023; Maini et al., 2024; Yu et al., 2023; Yao et al., 2024; Meng et al., 2022; Shin et al., 2020), particularly in response to evolving regulatory developments such as the GDPR (EU Commission, 2016). However, machine unlearning remains underdeveloped, with fundamental questions around its feasibility and reliability still unresolved (Xu et al., 2023). Critically, recent studies have highlighted limitations in current unlearning methods, noting their inability to guarantee data erasure and their potential to introduce new vulnerabilities (Hase et al., 2023; Chen et al., 2021).

Motivated by the need to understand how and where LLMs store memory as a prerequisite for effective machine unlearning, we systematically investigate transformer-based LLMs, with a particular emphasis on the often-overlooked role that attention layers play in underlying memory mechanisms.

Drawing on prior works in computational psycholinguistics that identified parallels between Transformer attention and cue-based retrieval theories of human sentence comprehension (Van Dyke & Lewis, 2003), we hypothesize that attention mechanism in Transformer implements memory-like functions analogous to those found in human cognition, encompassing three core processes: encoding, consolidation, and retrieval (Daumas et al., 2005; Guskjolen & Cembrowski, 2023). To

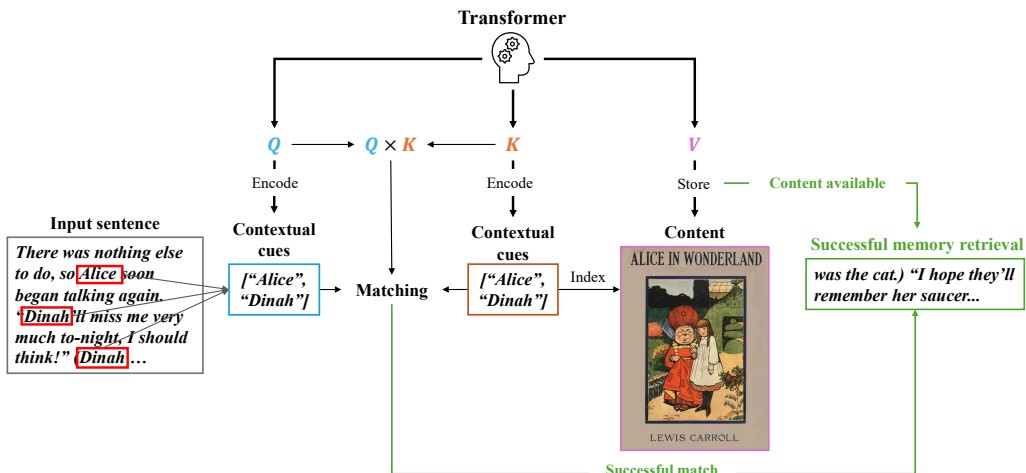

Figure 1: Illustration of the hypothesized memory retrieval process in Transformer models, grounded in cue-based retrieval theories and the Encoding Specificity Principle. The example depicts a successful retrieval event contingent on a strong cue–trace match and the availability of the relevant content.

evaluate this hypothesis, we analyze the roles of individual components of the attention mechanism in Transformer-based LLMs, an area for which grounded empirical evidence remains scarce.

Further guided by the Encoding Specificity Principle (ESP) (Tulving & Thomson, 1973), which posits that successful retrieval depends on the overlap between contextual cues present at encoding and those available at retrieval, we further proposed that in the context of LLMs for text generation tasks, these "contextual cues" are encoded as keywords by the attention mechanism. Figure 1 visualize our our hypotheses, illustrating the theorized roles of $Q$, $K$, and $V$ in memory retrieval.

In this paper, we use empirical results to prove our hypothesis, with key contributions include: (1) demonstration that Q, K, and V play distinct roles paralleling human memory—**Q as context encoder, K as trace memory index, and V as content store** (2) empirically confirming that **contextual cues are represented as keywords**; and (3) identifying **consistent, model-specific attention neurons activated by keywords**, suggesting a potential location for contextual memory.

## 2 RELATED WORK AND OBSERVATION

### 2.1 EXPLAINABLE ARTIFICIAL INTELLEGENCE (XAI)

Numerous studies in XAI have examined explainability in transformer LLMs using various techniques, including input perturbation (Fong & Vedaldi, 2017), surrogate models (Ribeiro et al., 2016; Guidotti et al., 2019; Leemann et al., 2025), gradient-based methods (Simonyan et al., 2014; Shrikumar et al., 2017; Lundberg & Lee, 2017), and layer-wise relevance propagation (LRP) (Bach et al., 2015; Achtibat et al., 2024a).

Existing studies start to put specific focus on attention layers for explainability (Bahdanau et al., 2014; Abnar & Zuidema, 2020). Hybrid approaches also exist, integrating attention mechanisms with other explainability methods to further enhance interpretability (Deiseroth et al., 2023; Achtibat et al., 2024a). The explanatory value of attention, however, remains debated (Jain & Wallace, 2019; Wiegreffe & Pinter, 2019), with recent studies suggesting that attention can provide alternative insights but does not fully address all goals of model explainability (Bastings & Filippova, 2020; Lopardo et al., 2024).

Different from other XAI approaches, human-interpretable explanations for XAI aim to clarify transformer models using metrics or reasoning comprehensible to humans. However, existing work has mainly focused on Feed-Forward Neural Network (FFNN) layers. Geva et al. (2021) showed that

FFNN layers act as Key-Value memories, with Key matrices matching recurring linguistic patterns. Geva et al. (2022) further demonstrated that Value matrices encode and promote high-level concepts during prediction, as reflected in token distribution shifts.

Beyond much of the conventional XAI literature, which seeks comprehensive explanations involving grammar, syntax, and semantics, our work specifically targets memory mechanisms in transformer models, emphasizing memory as one distinct and influential component that contributes to the accurate and human-like responses from LLMs.

## 2.2 ATTENTION MECHANISMS IN THE VIEW OF COMPUTATIONAL PSYCHOLINGUISTIC

In computational psycholinguistic field, Van Dyke & Lewis (2003) proposed the cue-based retrieval theories, which can be described as when memory retrieval is initiated, available retrieval cues are matched against all possible candidates via a direct and parallel matching process. In other words, retrieval mechanism in sentence comprehension is *content-addressable*. Since the introduction of the Transformer architecture (Vaswani et al., 2017), several studies in this field have drawn explicit parallels between the attention mechanism and cue-based retrieval theories (Yoshida et al., 2025; Ryu & Lewis, 2021; Timkey & Linzen, 2023; Oh & Schuler, 2022). Nevertheless, integration between XAI and computational psycholinguistics remains limited: XAI primarily seeks mechanistic explanations of model internals, whereas computational psycholinguistics leverages LMs to formalize and evaluate theories of human language processing. These divergent objectives have constrained meaningful exchange despite clear conceptual overlap.

Until recently, Gershman et al. (2025) articulated a key–value memory framework that bridges machine learning, psychology, and neuroscience, with implications for computational psycholinguistics. In this formulation, inputs are transformed into two distinct representations—keys and values—that are stored in memory: keys serve as memory indices enabling content-addressable access, while values encode the memory content. Memory retrieval proceeds by forming a query and matching it against the stored keys; the resulting matches identify the corresponding relevant memory contents in values. This formulation aligns with cue-based retrieval theories in psycholinguistics and with the Transformer attention mechanism. To support this view, the authors introduce two toy models that emulate self-attention (with Q, K, and V) and feedforward neural network. These models show that keys and values are represented distinctively to align with their respective roles in memory retrieval, and that forgetting can reflect retrieval failure even when memory traces persist-an analogue of the tip-of-the-tongue phenomenon in human memory (Freedman & Landauer, 2014).

In this paper, we move beyond toy settings by conducting experiments on multiple LLMs to test for key–value memory in their attention mechanisms. We treat the Encoding Specificity Principle (ESP; (Tulving & Thomson, 1973)) as a unifying foundation for both cue-based retrieval and key–value memory: retrieval succeeds to the extent that retrieval cues overlap with features encoded at memorization step. We use ESP to guide our hypothesis of Transformer attention functionalities, which can be formalized as:

**Hypothesis 1 .**   Q encodes retrieval cues, K indexes candidate traces by those cues, and V stores retrievable content.

## 2.3 OBSERVATION

Having outlined theoretical motivations from XAI and computational psycholinguistics, we now present observations of self-attention that motivate our hypothesis. Sukhbaatar et al. (2019) noted a close similarity between self-attention and FFNN, as both rely on dot-product–based linear projections. Building on this and recent FFNN explainability results (Geva et al., 2021; 2022), which characterize FF layers as performing pattern matching over learned keys and promoting associated concepts via their values, we ask whether attention layers enact an analogous process. Following the ESP-based mapping introduced above ($Q$ as retrieval cues, $K$ as indices, $V$ as stored content), we treat attention weights as content-addressable matching and further hypothesize:

**Hypothesis 2 .**   The retrieval cues are instantiated as salient lexical tokens ("keywords") tied to the relevant memory.

Hypothesis 2 is consistent with findings by Eldan & Russinovich (2023), who propose an approximate unlearning method that replaces topic-specific keywords with generic placeholders. Scrubbing these lexical cues markedly impairs the model's ability to retrieve facts about the target topic. Despite the collateral degradation, the results indicate that such keywords function as critical retrieval cues in LLMs.

From a mathematical standpoint, the attention mechanism in language models was initially proposed as a similarity measurement (Bahdanau et al., 2014), due to the utilization of dot product calculation, where similar vectors yield higher values. Therefore, the implementation of self-attention with $Q$, $K$, and $V$ (Vaswani et al., 2017) closely resembles a form of content-based lookup, where each components perform a dot product transformation on the initial input embeddings to create distinct representations for their unique roles aligning with Hypothesis 1. This view is shared by many past papers in machine learning field or greatly hinted at (Tay et al., 2021; Roy et al., 2021; Rohekar et al., 2023). In the equation for self-attention:

## 3 METHODOLOGY

In this section, we describe our two experiments designed to empirically prove Hypothesis 1 and 2. For both experiments, we employ six different auto-regressive (decoder-only) LLMs of varying sizes and structures: **Llama 2-7b**, **Llama 2-13b** (Touvron et al., 2023), **Llama 3.1-8b** (Grattafiori et al., 2024), **Olmo 2-1124-13b** (OLMo et al., 2025), **Qwen 2.5-14b** (Qwen et al., 2025), **Phi-4** (Abdin et al., 2024) and **GPT-Neox-20b** (Black et al., 2022).

### 3.1 EXPERIMENT 1: EMPIRICAL VALIDATION OF HYPOTHESIS 1 WITH ATTENTION SWAPPING

**Experiment description:** Self-attention in Transformer is calculated as:

$$\text{Attention} = \text{softmax}\left(\frac{QK^T}{\sqrt{d_k}}\right) V \tag{1}$$

According to the ESP and the key-value memory framework (Section 2.2), memory retrieval relies on two processes: Content-addressable matching ($QK^T$) and the availability of stored content ($V$). Depending on the successful of each process, we can have three different cases when models retrieve memory. **Case one**: matching works and stored content is present, representing successful memory retrieval. **Case two**: matching fails and stored content is present, representing normal forgetting or the tip-of-the-tongue phenomenon. **Case three**: matching works and stored content is not present, but what might this represents ? Intuitively, we conjecture that case three represents hallucination, where models identify a match but produce incorrect content. To this end, we conducted an experiment in which we interchange the $Q$, $K$, and $V$ projections—either individually or in pairwise combinations—across prompts to reproduce **case two** and **three** and examine whether the corresponding phenomena emerge, hence empirically validating Hypothesis 1

**Dataset:** Because not all prompts permit objective quantification of hallucination in our experiment, we therefore employ the Counterfactual dataset (Meng et al., 2022) - a dataset comprises of factual and counterfactual examples that are structurally similar but contextually distinct, thereby eliciting different target outputs from the model that can be quantified.

Swapping is conducted on a pairwise basis. For each factual example in the dataset, we find all suitable counterfactual examples to create our swapping pair of prompts. Suitable counterfactual examples must have the same length (after tokenized) as their corresponding factual example for precise swapping of $Q$, $K$, and $V$ projections. So, for a factual example $x_f$, its counterfactual example $x_{cf}$, and their corresponding $Q/K/V$, our swapping can be formulated as:

$$\text{Swapped Attention K}(x_f) = \text{softmax}\left(\frac{Q_{x_f} K_{x_{cf}}{}^T}{\sqrt{d_k}}\right) V_{x_f} \tag{2}$$

Note that our swapping only occurs for the input prompts, not the subsequent generated text. Thus, the projections computed for $x_{cf}$ are not imposed on $x_f$ after processing the input context. We allow

LLMs to revert back to their original $Q/K/V$ projections after first token is generated. This design constitutes a less intrusive intervention than swapping for the entirety of generation. Our targets for swapping include $K$, $V$, and $KV$. Swapping of $Q$ or any other combinations are not required as they are implied by our three chosen targets (e.g., swapping $Q$ is the same as swapping $KV$).

**Metrics:** Metrics used for this experiment include accuracy (for both factual and counterfactual labels), which is computed as first-word exact match to measure hallucination effect. Additionally, we compute the $\Delta logit$ and the perplexity overhead to assess the extent to which the swapping procedure induces hallucination. Both metrics are defined as differences between the model's outputs under the original and swapped conditions.

## 3.2 EXPERIMENT 2: EMPIRICAL VALIDATION OF HYPOTHESIS 2 WITH K MATRIX PERTURBATION

**Experiment description:** Both $Q$ and $K$ encode retrieval cues that support content-addressable matching; hence, either component could, in principle, be manipulated to probe cue representations in a sentence-comprehension setting. However, because queries are evaluated against keys (rather than the reverse), we hypothesize that $K$ is the more informative target for intervention.

**Dataset:** To evaluate Hypothesis 2, the experiment requires datasets with clearly interpretable contexts and recognizable by unambiguous lexical cues. Accordingly, we use long-form book text drawn from publicly available corpora - Project Gutenberg (n.d.). The dataset is perfect for this experiment as each book has its own unique storyline, characters, and places.

Raw text of each book is processed with input/label pairs generated by a sliding window technique (step size 30, input: 512 tokens, output: 40 tokens) to create a dataset $\mathbb{D}_i$. Input and label combined form complete sentence(s) in the books. Furthermore, to best represent the model's memory, we additionally filter for verbatim examples using ROUGE-L Recall from the ROUGE suite (Lin, 2004), a metric widely used to measure overlap between generated texts, and commonly adopted/built upon for evaluating unlearning effectiveness (Jang et al., 2023; Carlini et al., 2023). The final result yield $\mathbb{A}_i$ for $i \in 1, \ldots, n$, where $n$ is the number of books:

$$\mathbb{A}_i = \{a_1, a_2, \ldots, a_n\}, \quad where : \text{ROUGE-L Recall}(a_j) = 1, \forall a_j \in \mathbb{D}_i \tag{3}$$

Note that the set of $\mathbb{A}_i$ for each model is different since not all models share the same memory about the same book. (The specific list of books used for each model, along with their ROUGE-L Recall sample sizes, can be found in Appendix B.)

Following Eldan & Russinovich (2023), we use ChatGPT-4o to identify *anchored terms* - $AT_i$ (each dataset $\mathbb{A}_i$ has a corresponding $AT_i$) as keywords for each book to test our hypothesis (see Appendix A for our exact prompt). Let $x = w_1, \ldots, w_n$ denote an input, where each $w$ is a word in $x$. We extract the $K$ projected values (from Equation 1) for all $w \in AT_i$ at a given layer $l$ and head $h$. The dataset-level coefficient for each layer-head pair is the mean across all inputs:

$$\text{m}_{l-h}^{\mathbb{A}_i} = \frac{1}{n} \sum_{i=1}^{n} \left\{ e_j W_{l-h}^K \mid w_j \in AT_i \right\} \tag{4}$$

where $n$ is the total number of inputs in dataset $\mathbb{A}_i$ and $e$ is the embedding vector for word $w$. To identify the most significant layers or attention heads, we aggregate and average these scores across heads or layers. This approach also enables direct ranking of hidden units (layer-head-dimension triplets).

Our method of identifying keywords is based on observed activation patterns of top neurons in response to GPT-4o-generated keywords (Equation 4). Keywords are identified by examining top words weighted by the learned weight matrix $W^K$. To produce the final list of keywords for a dataset, we aggregate the scores of top words across all inputs:

$$S_{\mathbb{A}_i}(t) = \sum_{x \in \mathbb{A}_i} \left\{ \frac{1}{N_x(w)} \sum_{j=1}^{N_x(t)} \text{score}_{x,j}(w) \right\} \tag{5}$$

where $N_x(w)$ is the number of occurrences of word $w$ in $x$; $\text{score}_{x,j}(t)$ refers to the sum of key projection scores $\tau W^K$ over all sub-word tokens $\tau$ (since LLMs utilize sub-word tokenization) that constitute the $j$-th occurrence of $w$ in input $x$.

We empirically assess the effect of our identified keywords on model memory, benchmarking against keywords produced by GPT-4o and by a state-of-the-art attention-informed XAI method, Layer-wise Relevance Propagation eXplains Transformers (LXT) (Achtibat et al., 2024b). We fix the budget at 20 keywords per method. For LXT, we collect the top-relevance tokens per input and aggregate them as in Equation 5 to obtain a top-keyword list for each book dataset. We exclude the first and last tokens for all inputs because LXT systematically assigns the highest and lowest relevance scores to the final and initial tokens, respectively, which would otherwise yield keyword lists dominated by end-of-input tokens. This behavior highlights a difference in goals between our work and prior XAI approaches, as discussed at the end of Section 2.1. Nevertheless, we report LXT results to contextualize and further support our findings.

A simple perturbation at $K$ for identified keywords by setting their projected values to 0:

$$\forall w \in x, \quad w \in \text{AT}i \implies eW_\alpha^K = 0 \tag{6}$$

where $\alpha$ can be specific layers, heads, layer-head pairs, or layer-head-dimension depending on the desired level of granualarities. This perturbation is equivalent to none of the words in the input attending to identified keywords at selected neurons. We focus on perturbing specific attention heads, based on the intuition that certain heads may be responsible for memory mechanisms, as prior work has shown that individual heads often serve distinct functions (Voita et al., 2019; Clark et al., 2019; Vig, 2019). We compare perturbation outcomes against both the unperturbed baseline and perturbations applied to randomly selected non-keyword tokens. For each input, we sample as many random tokens as there are identified keywords. Because interventions on K are expected to affect model behavior to some extent, the randomized control estimates the impact of indiscriminate perturbations. If targeting extracted keywords produces larger deviations from baseline than targeting random tokens, this indicates that the identified terms function as retrieval cues for content-addressable matching.

Top heads identified by Equation 5 are selected for perturbation. To ensure proportionality across model sizes, we select 2 heads for **Llama 2-7b**, 3 for **Llama 2-13b**, and 4 for **GPT-Neox-20b** models. For **Llama 3.1-8b**, **Qwen 2.5-14b**, and **Phi-4** (14b), we select 1, 2, and 2 respectively due to their architectural design that employs Multi-Query Attention (MQA) or Group Query Attention (GQA), where attention heads are grouped together for faster training. We exclude the first attention head if it appears among a model's top-ranked heads, replacing it with the next-ranked head. This choice is motivated by prior work showing that the first head functions as an induction head that facilitates information flow to subsequent heads rather than supporting memory-specific computations (Muşat, 2025). The only exception to the rest is **Olmo 2-1124-13b**, where we perturb for all heads across all layers, and more will be discussed in Section 4.2 about this decision. Metrics used for evaluation are: ROUGE-L Recall, Perplexity, BERTScore (Zhang* et al., 2020), Repetition rate, and MAUVE (Pillutla et al., 2023). To facilitate comparison, all metrics are normalized to the unperturbed baseline to emphasize proportional changes.

**Metrics:** We assess impact of keyword to memory recall ability with ROUGE-L Recall and BERTScore, the latter leveraging contextual embeddings to quantify semantic similarity. General generation capabilities under perturbation is evaluated using perplexity, repetition rate, and MAUVE. MAUVE is an unsupervised evaluation framework that quantifies the distributional similarity between model-generated and human-written text. Repetition rate is our custom metric that measures the repetitiveness of generated text based on n-gram overlapping rate:

$$\text{repetition rate} = 1 - \frac{|\text{unique n-gram}|}{|\text{total n-gram}|} \tag{7}$$

## 4 RESULTS

### 4.1 EXPERIMENT 1: ATTENTION SWAPPING

**Swapping $V$ significantly increases hallucinations.** Figure 2 shows the results for **Experiment 1**. To our surprise, swapping $V$ alone induces LLMs to hallucinate and answer as if they were prompted with samples from counterfactual set for minimum 50% in **Olmo-2** and maximum 90% in **Qwen2.5** (Green bars in the upper-right subfigure). Across models, the average perplexity overhead remains minimal, and the mean $\Delta logit$ is positive for most models; the exceptions are **GPT-NeoX-20B** and

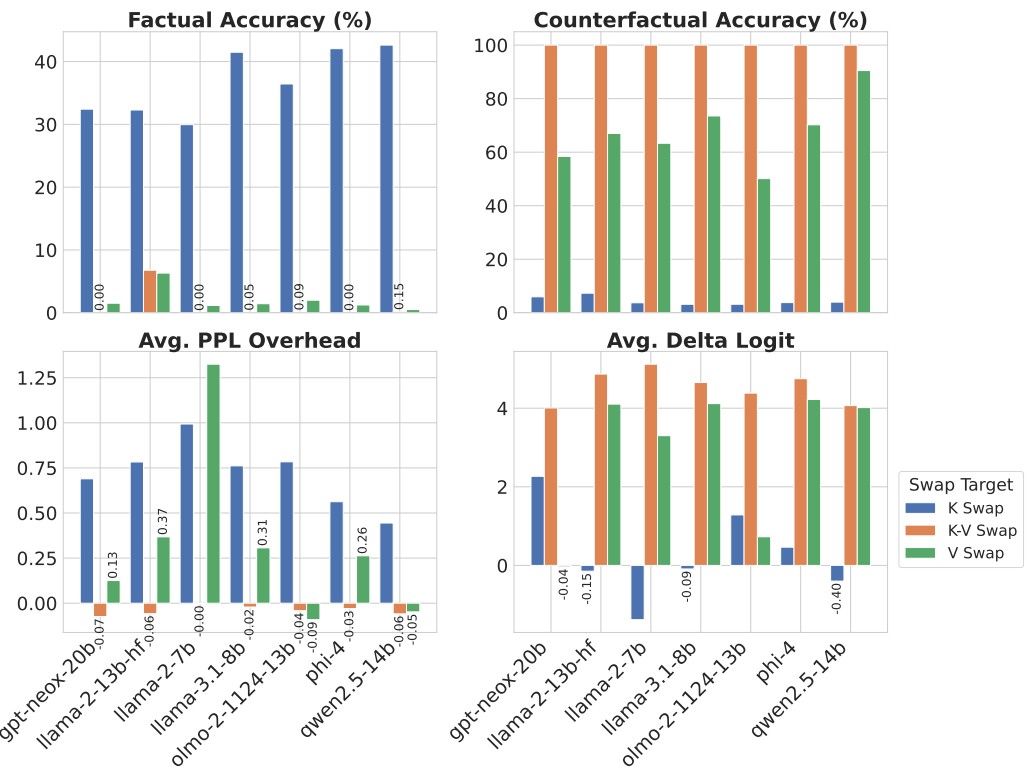

Figure 2: $QKV$ swapping experiment results.

**OLMo-2**, which also exhibit the lowest counterfactual accuracy. Overall, these findings indicate a clean replacement of knowledge achieved by swapping $V$. The result aligns with key-value memory framework and strengthen our view in Hypothesis 1, where $V$ plays the role of content storage.

**Swapping both $K$ and $V$ induces LLMs to hallucinate for 100% of input factual prompts across all experimented models** (Orange bars in the upper-right subfigure). Recall that Q and K are hypothesized to implement content-addressable matching via the dot-product calculation (similarity measurement - see Section 2.3). When $K$ is replaced with $K_{cf}$, the model computes attention between the factual queries ($Q_f$) and counterfactual keys ($K_{cf}$), effectively performing nearest-neighbor search in the counterfactual key space. This steers the address toward counterfactual memory slots; because the corresponding values ($V_{cf}$) are also supplied, the retrieved content is counterfactual, yielding systematic hallucination. This also explains the improvement (though minimum) in perplexity and the consistent positive $\Delta logit$ when swapping both $K$ and $V$.

On the other hand, **swapping $K$ only does not induce hallucination but only hinder model's ability to recall factual knowledge** where the factual accuracy across models is around 30%-40%. The results in this swapping setting strengthen the views from computational psycholinguistics and psychology about attention mechanism. Specifically, with $V$ intact, the memory content must be available but models "forget" due to retrieval failure.

### 4.2 EXPERIMENT 2: $K$ MATRIX PERTURBATION

**Consistent top neurons activated when prompting keywords**. Recall that each book dataset is associated with a unique set of GPT-4o-generated keywords. Figure 3 ranks the neurons (layer–head–dimension triplets) most activated by keywords. For each model, we compute an average reciprocal rank score across the book datasets, weighting rank $r$ by $1/r$; thus, the second-ranked neuron contributes half the score of the first. The results reveal, for each model, a single dominant neuron whose score significantly exceeds that of the second-ranked neuron. The same observation

Table 1: Keyword lists by extraction method for Alice's adventure in wonderland book.

| GPT-4o Generated | LXT | Our method |
|---|---|---|
| alice, rabbit-hole, sister, bank, daisies, white rabbit, waistcoat-pocket, orange marmalade, dinah, schoolroom, new zealand, australia, latitude, longitude, cheshire, duchess, caucus-race, queen, mock turtle, lobster quadrille | be, alice, might, an, had, thought, in, time, her, moment, by, said, barrowful, this, could, atom, she, as, what, down | alice, dormouse, whiskers, sneezing, mushroom, hastily, caterpillar, angrily, hurry, aloud, dinah, melancholy, queer, timidly, lefthand, nursing, frightened, fountains, rabbit, sleepy |

is true for higher levels of granularity (see Appendix C for a detailed ranking of neurons across available book dataset for each model). The results indicate the presence of unique neurons within each LLM that show strong evidence of encoding/consolidating contextual memory in the form of keywords.

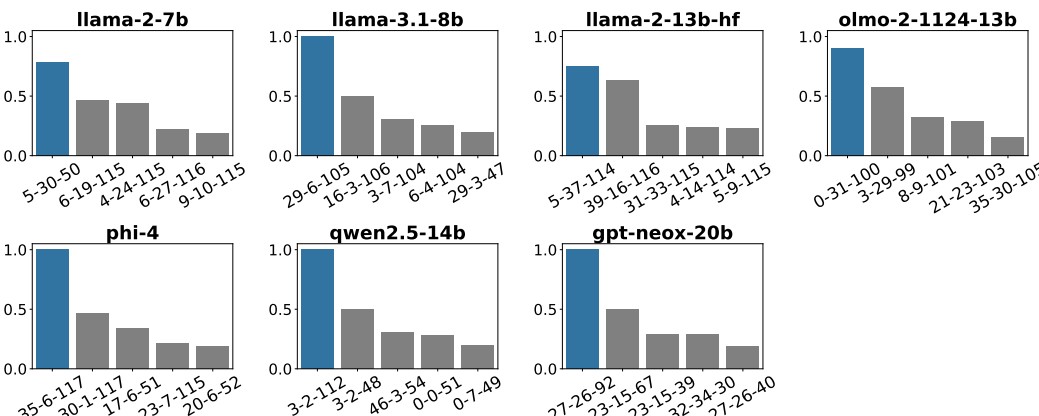

Figure 3: Mean reciprocal rank of layer-head-dimension for each model across its respective book datasets.

**Perturbing linearly projected keywords at *Key matrix weights* impairs memory retrieval**. We observe that our generated keywords can be easily associated with the relevant topic and semantically comparable to the ones generated by GPT-4o. Table 1 shows a comparison of keywords extracted by different methods for **Llama-2-7b**.

Figure 4 presents the results of the perturbation experiments, where each column shows the result of one metric, upper and lower row show the between perturbation of extracted keywords and randomly selected words respectively. We also show the standard deviation band, with the exception of MAUVE. Because MAUVE requires a large sample size for reliable estimation, we compute it by pooling all inputs across datasets $\mathbb{A}_i$ available to each model, rather than computing it per input; consequently, standard deviation band is not available for MAUVE.

Perturbations using LXT-extracted keywords yield the largest reduction in memorization as measured by ROUGE-L recall and BERT-Score, followed by our method and the GPT-4o–generated method. However, MAUVE and the repetition rate degrade markedly under LXT relative to the baseline and the other methods. The standard-deviation band for LXT on the repetition-rate metric is especially wide—particularly for the Llama-2 family—whose lower bound approaches zero. Because our method perturbs every keyword detected in an input, the perturbation size can become large when the extracted list contains common, high-frequency words; the scope is therefore input-dependent. This suggests that XAI methods such as LXT tend to identify tokens predictive of the model's output but do not reliably isolate content-specific, context-bearing keywords.

Compared with random word selection (second row), all methods achieve greater reductions in memorization as measured by ROUGE-L recall, except for **Llama-2-7B** and **Llama-2-13B**, for which ROUGE-L recall is nearly indistinguishable across the three keyword-extraction approaches.

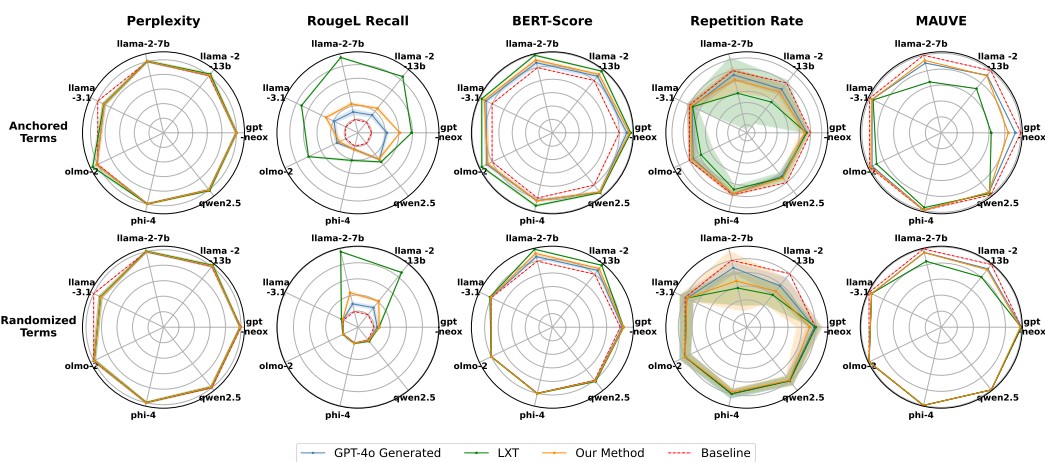

Figure 4: Radar graphs showingh average overall performance for all evaluated models when perturbed with different methods (individual book results can be found in Appendix D). All results are normalized to show higher is better.

We argue that this behavior, unique to Llama 2 family models is due to the small vocabulary size of 32.000 compared to its successor **Llama 3.1** with 128.256 (about four times larger).

Crucially, we find that an extreme case of using keywords as contextual cues for content-addressable matching in **Olmo 2-1124-13b**. Specifically, perturbing at ALL attention heads for keywords sharply reduces ROUGE-L Recall and BERTScore, while random word perturbations have minimal effect. This suggests that the model relies almost exclusively on keywords for memory retrieval, indicating minimal feature superposition for memory features. In summary, the results validate Hypothesis 2 by showing that neglecting keywords in attention layers significantly impairs the memory retrieval capabilities of LLMs. Presenting retrieval cues used in memory retrieval process of Transformers as salient lexical tokens. Additionally, we identify specific neurons encode keyword-related memory, enabling dynamic keyword extraction for a given topic.

## 5 CONCLUSION

We propose that transformer-based LLMs recall memory in accordance with the Encoding Specificity Principle. Through extensive empirical analysis, we demonstrate that the internal mechanisms of attention align with human memory processes during sentence comprehension, consistent with cue-based retrieval theories and the key–value memory framework.

We show that the Transformer's attention mechanism leverages the key-projection $(K)$ weights to index memory traces via cues encoded in salient lexical tokens, thereby facilitating retrieval. We further identify model-specific units whose activations are selectively driven by these keywords, suggesting a role as context-addressable memory locations. Building on these results, we present a method to extract the keywords that index a remembered context, thereby enabling downstream applications such as unlearning.

**Limitation**. Our perturbation method offers a minimally invasive, context-specific approach to machine unlearning. Its main limitation is naivety: the choices of top neurons and the number of targeted keywords remain largely arbitrary. Future work should optimize selection criteria and modification strategies, moving beyond simple zeroing.

On the other hand, our method of extracting keywords also has limitations that originate from compound words or terms made up of more than 1 word. For example, "White rabbit" is a better keyword than simply "rabbit", but our method cannot treat "white" and "rabbit" together as a single term. As a result, the list of keywords extracted by our method does not fully capture the ideal set of contextual cues.

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

## A GPT4-O PROMPT TO EXTRACT ANCHORED TERMS

```
[You are a helpful assistant. A long passage of text will be provided to you.
Your task is to extract a list of 20 (in total) expressions, names or
entities which are idiosyncratic to the text (try your best to keep in
between 1-2 words, THE SHORTER THE BETTER). Please extract exactly how
they appear in the text but in lowercase.]
```

Table 2: 1.0 ROUGE-L Recall samples count for all models.

| Model | No. Samples ($AT$) |
|---|---|
| **llama-2-13b** | |
| Alice's Adventures in Wonderland (Carroll, 1865) | 278 |
| Dracula (Stoker, 1897) | 218 |
| Frankenstein (Shelley, 1818) | 513 |
| The Great Gatsby (Fitzgerald, 1925) | 89 |
| Moby-Dick (Melville, 1851) | 41 |
| Pride and Prejudice (Austen, 1813) | 327 |
| The Adventures of Sherlock Holmes (Doyle, 1892) | 583 |
| **llama-2-7b** | |
| Alice's Adventures in Wonderland (Carroll, 1865) | 78 |
| Frankenstein (Shelley, 1818) | 51 |
| The Great Gatsby (Fitzgerald, 1925) | 43 |
| Pride and Prejudice (Austen, 1813) | 83 |
| The Adventures of Sherlock Holmes (Doyle, 1892) | 62 |
| **olmo-2-1124-13b** | |
| Alice's Adventures in Wonderland (Carroll, 1865) | 296 |
| The Great Gatsby (Fitzgerald, 1925) | 38 |
| Moby-Dick (Melville, 1851) | 66 |
| The Odyssey (Homer, 2008) | 84 |
| Peter Pan (Barrie, 1920) | 296 |
| The Adventures of Tom Sawyer (Twain, 1876) | 743 |
| The Wonderful Wizard of Oz (Baum, 1900) | 528 |
| **gpt-neox-20b** | |
| Alice's Adventures in Wonderland (Carroll, 1865) | 38 |
| Frankenstein (Shelley, 1818) | 113 |
| Moby-Dick (Melville, 1851) | 38 |
| The Adventures of Sherlock Holmes (Doyle, 1892) | 51 |
| **phi-4** | |
| The Great Gatsby (Fitzgerald, 1925) | 39 |
| Pride and Prejudice (Austen, 1813) | 86 |
| Frankenstein (Shelley, 1818) | 214 |
| Moby-Dick (Melville, 1851) | 36 |
| Alice's Adventures in Wonderland (Carroll, 1865) | 50 |
| **llama-3.1-8b** | |
| The Adventures of Tom Sawyer (Twain, 1876) | 82 |
| Wuthering Heights (Brontë, 1847) | 44 |
| Alice's Adventures in Wonderland (Carroll, 1865) | 811 |
| Moby-Dick (Melville, 1851) | 106 |
| The Great Gatsby (Fitzgerald, 1925) | 167 |
| Dracula (Stoker, 1897) | 347 |
| The Odyssey (Homer, 2008) | 48 |
| The Adventures of Sherlock Holmes (Doyle, 1892) | 253 |
| Frankenstein (Shelley, 1818) | 466 |
| Pride and Prejudice (Austen, 1813) | 458 |
| The Wonderful Wizard of Oz (Baum, 1900) | 54 |
| Oliver Twist (Dickens, 1838) | 37 |
| **qwen2.5-14b** | |
| Alice's Adventures in Wonderland (Carroll, 1865) | 120 |
| Pride and Prejudice (Austen, 1813) | 40 |
| The Great Gatsby (Fitzgerald, 1925) | 31 |
| Frankenstein (Shelley, 1818) | 36 |
| Moby-Dick (Melville, 1851) | 34 |

## B    MODELS AND THEIR DATASETS

See Table 2

## C    TOP MEMORY COEFFICIENT NEURONS FOR ALL OTHER MODELS

Figure 5, Figure 6, Figure 7, Figure 8, Figure 9, Figure 10, and Figure 11 show the top memory coefficient neurons for **Llama 2-7b**, **Llama 2-13b**, **Olmo 2-13b**, **GPT Neox 20b**, **Llama 3.1-8b**, **Phi 4**, and **Qwen 2.5-14b** respectively.

## D    EVALUATION OF INDIVIDUAL BOOKS FOR ALL MODELS

Figure 12: **Llama 2-7b**, Figure 13: **Llama 2-13b**, Figure 14: **Olmo 2-13b**, Figure 15: **GPT Neox-20b**, Figure 16: **Llama 3.1-8b**, Figure 17: **Phi 4**, and Figure 18: **Qwen 2.5-14b**.

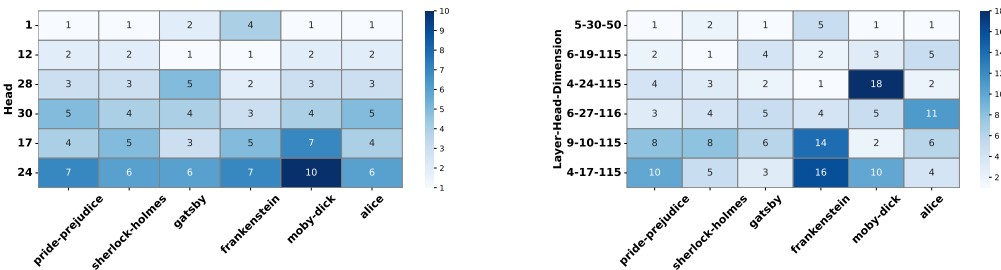

Figure 5: Top Memory Coefficient for **Llama 2-7b**. (**Left**): Top attention heads (**Right**): Top layer-head-dimension triplets

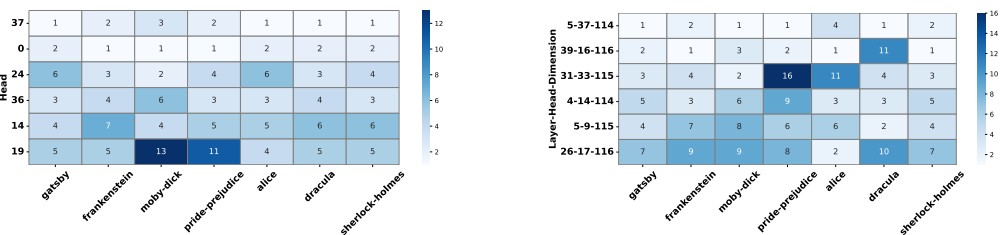

Figure 6: Top Memory Coefficient for **Llama 2-13b**. (**Left**): Top attention heads (**Right**): Top layer-head-dimension triplets.

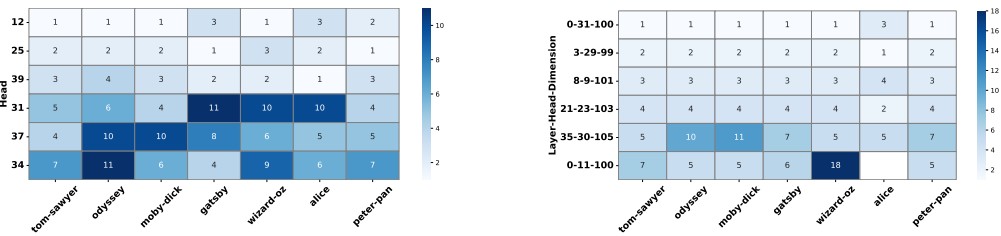

Figure 7: Top Memory Coefficient for **Olmo 2-13b**. (**Left**): Top attention heads (**Right**): Top layer-head-dimension triplets.

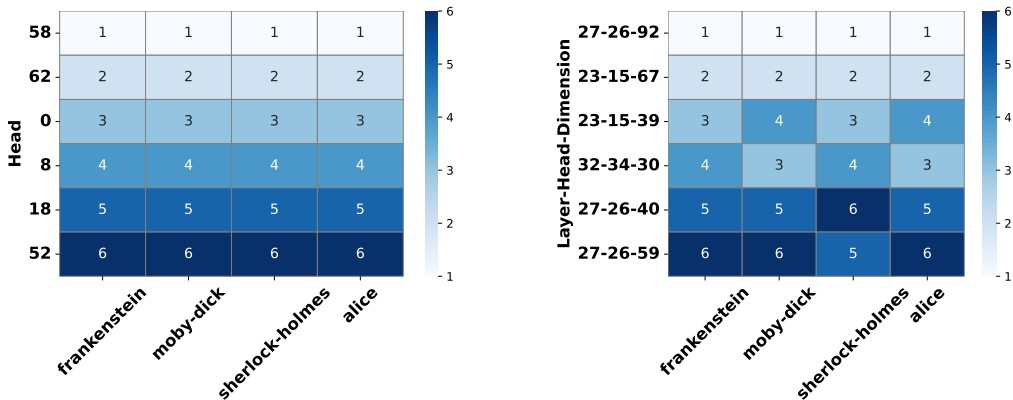

Figure 8: Top Memory Coefficient for **GPT-Neox**. (**Left**): Top attention heads (**Right**): Top layer-head-dimension triplets.

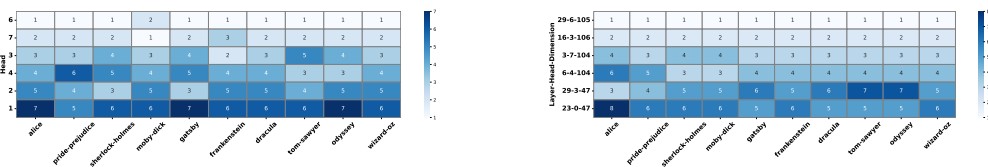

Figure 9: Top Memory Coefficient for **Llama 3.1-8b**. (**Left**): Top attention heads (**Right**): Top layer-head-dimension triplets

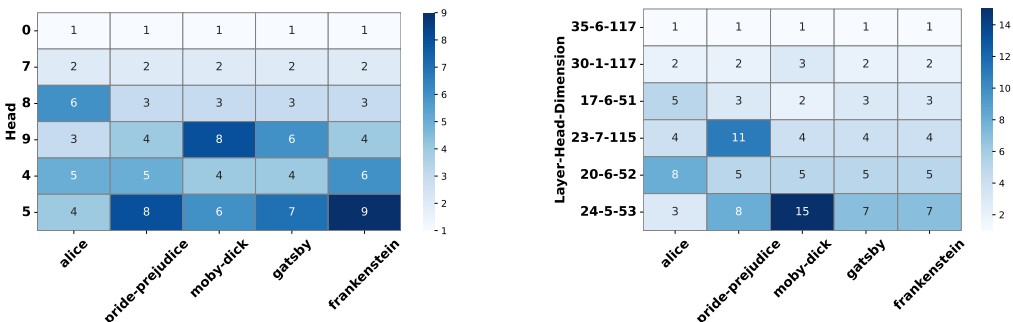

Figure 10: Top Memory Coefficient for **Phi 4**. (**Left**): Top attention heads (**Right**): Top layer-head-dimension triplets

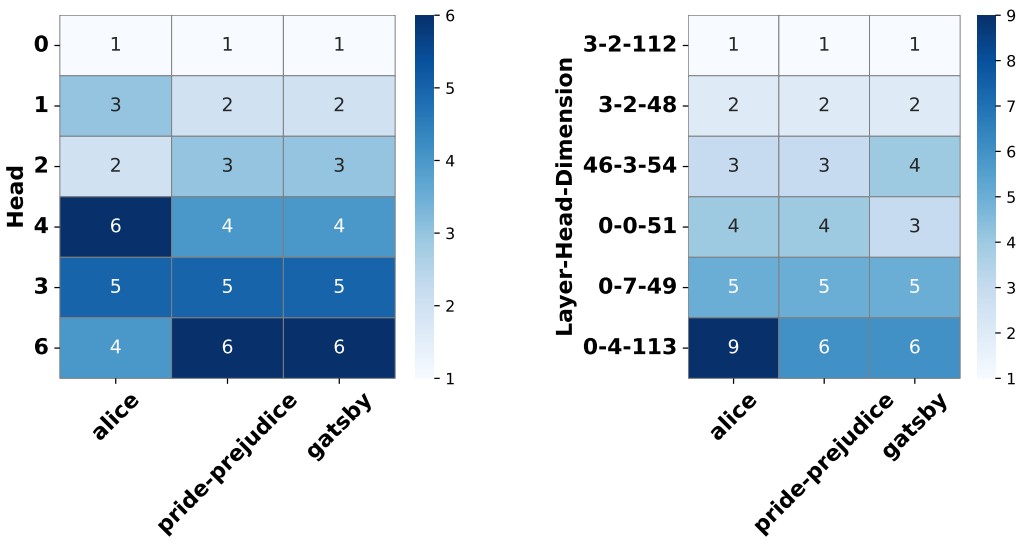

Figure 11: Top Memory Coefficient for **Qwen 2.5-14b**. (**Left**): Top attention heads (**Right**): Top layer-head-dimension triplets

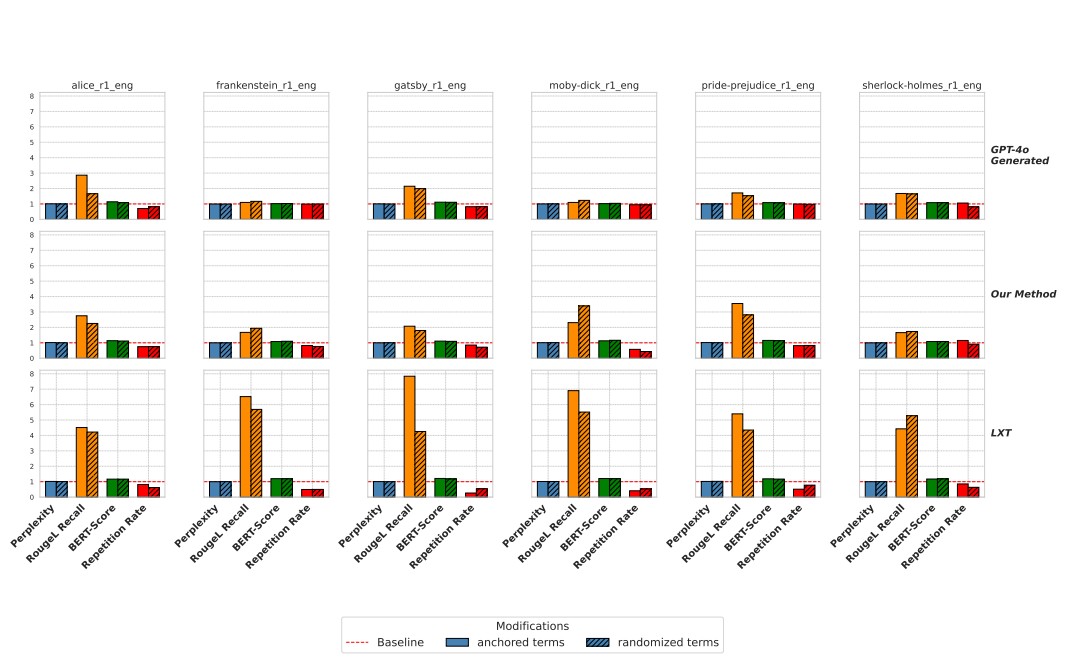

Figure 12: Evaluation on individual books for Llama 2-7b.

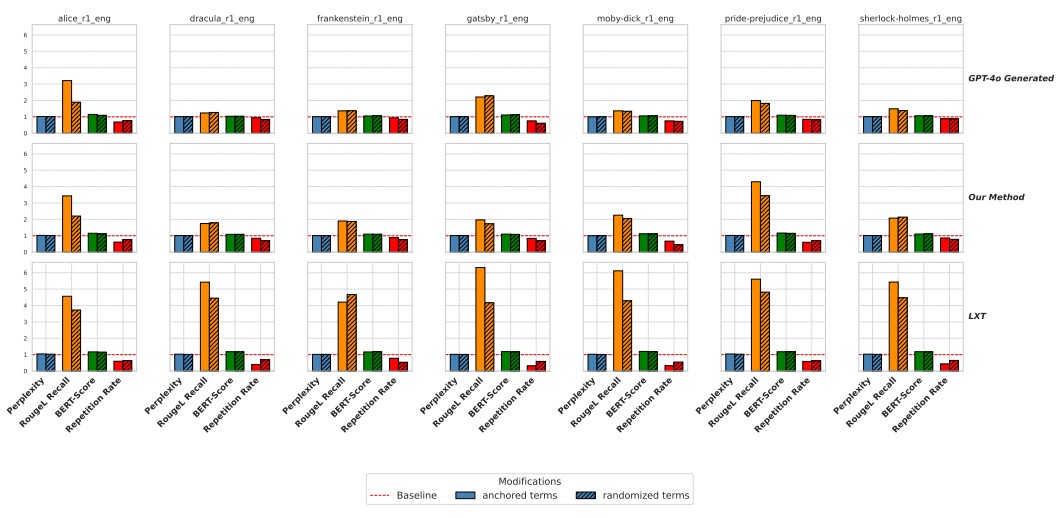

Figure 13: Evaluation on individual books for Llama 2-13b.

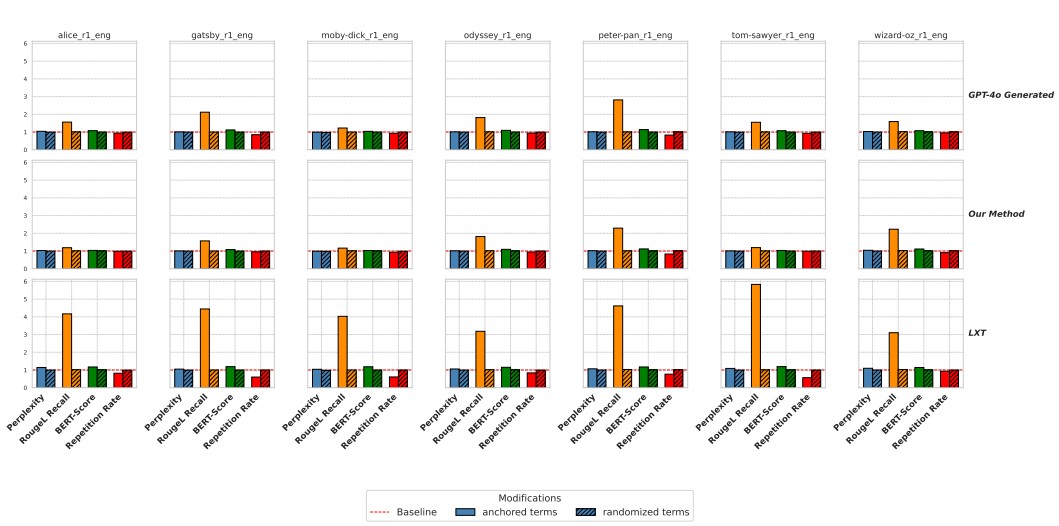

Figure 14: Evaluation on individual books for Olmo 2-13b.

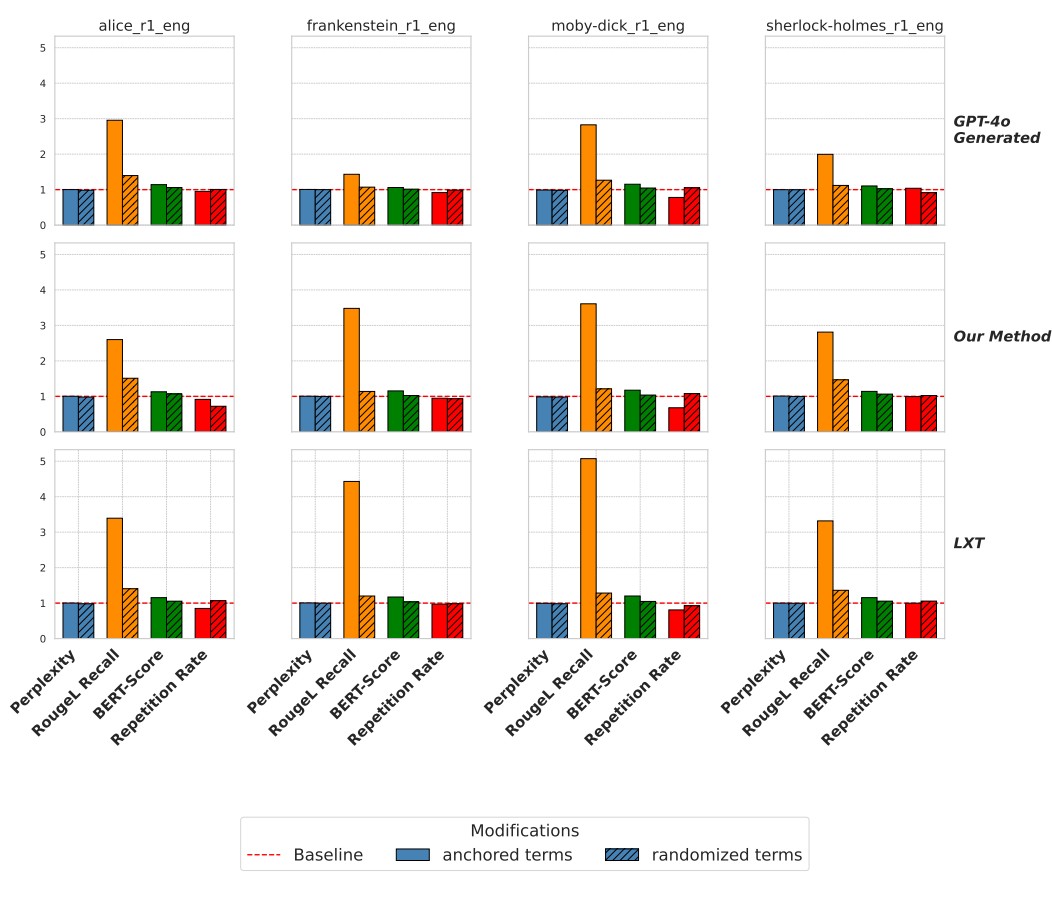

Figure 15: Evaluation on individual books for GPT Neox-20b.

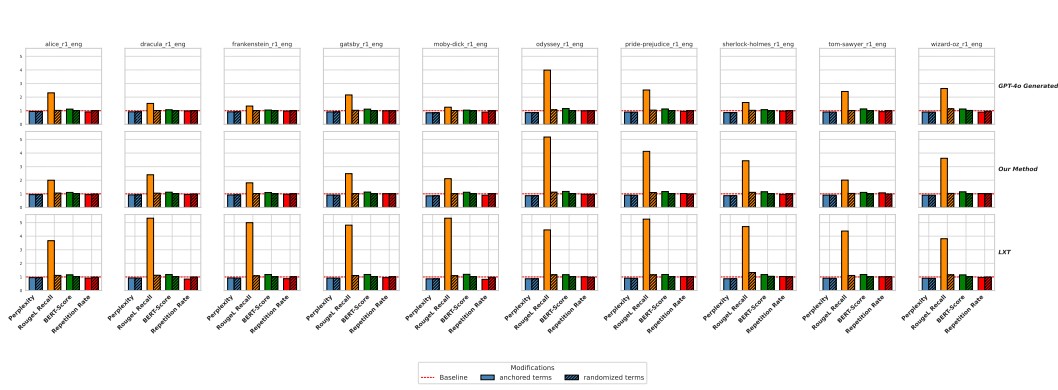

Figure 16: Evaluation on individual books for Llama 3.1.

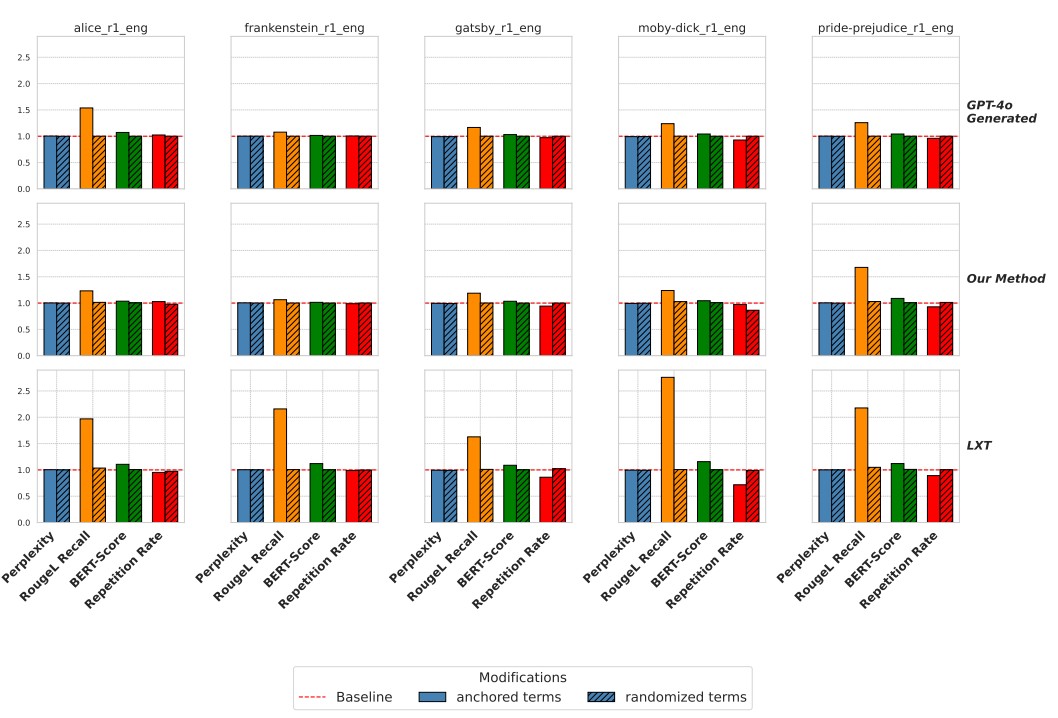

Figure 17: Evaluation on individual books for Phi 4.

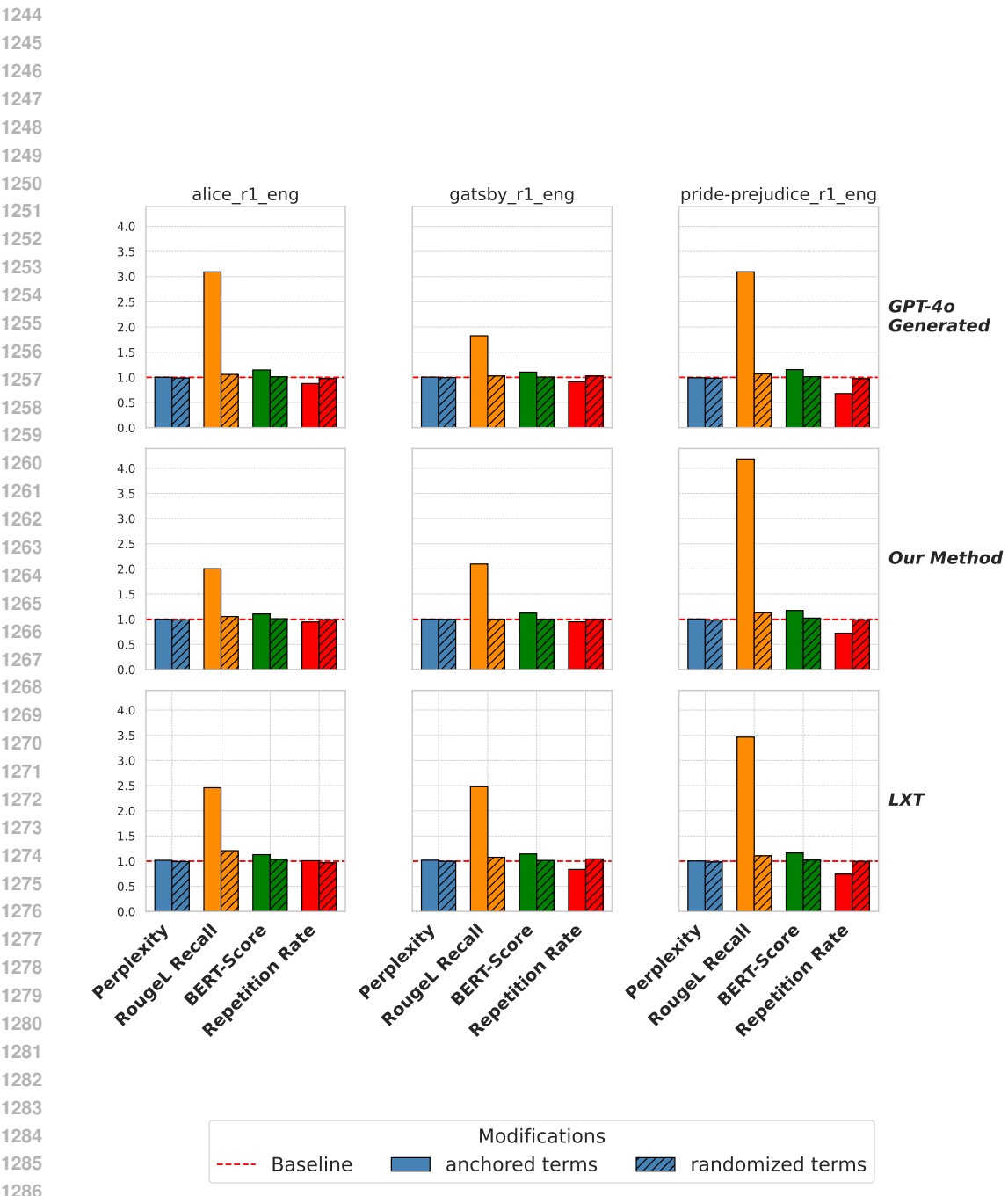

Figure 18: Evaluation on individual books for Qwen 2.5-14b.

