# OpenReview forum: "Memory Retrieval in Transformers:  Insights from the Encoding Specificity Principle"
_ICLR.cc/2026/Conference — Submitted to ICLR 2026_

### Official Review · Reviewer_ei5w · 2025-10-28

**Soundness:** 3
**Presentation:** 3
**Contribution:** 3
**Rating:** 4
**Confidence:** 4

**Summary:**

This paper investigates how transformer attention layers function as memory retrieval mechanisms, grounding the analysis in the Encoding Specificity Principle (ESP) from cognitive psychology. The authors argue that attention performs cue-based retrieval analogous to human memory.

**Strengths:**

Conceptual: Establishes a cognitive analogy between human cue-based memory retrieval and transformer attention under the Encoding Specificity Principle.

Empirical: Demonstrates that Q, K, V implement distinct cognitive-like roles—context encoding, memory indexing, and content storage—validated through controlled swapping and perturbation experiments.

Mechanistic: Identifies specific attention-layer neurons encoding “keywords” that act as retrieval cues, offering a concrete locus for contextual memory inside LLMs.

Applied: Suggests practical applications for machine unlearning and privacy-aware data removal, by targeting or suppressing these keyword-linked neurons to erase specific memories.

**Weaknesses:**

This paper is conceptually interesting but offers limited substantive innovation. The proposed connection between transformer attention and the Encoding Specificity Principle is largely a loose analogy rather than a formal theoretical contribution. The authors do not provide a rigorous mathematical formulation or define concrete quantitative measures such as memory retrieval efficiency or cue–content overlap. As a result, the findings are primarily descriptive phenomena rather than statistically grounded or systematically analyzed results.

Furthermore, the paper lacks stronger visualization or causal interpretability analysis. It does not present attention-head–level retrieval trajectories or activation dynamics that could substantiate the proposed analogy. Incorporating feature visualization or attention circuit tracing would make the conclusions considerably more convincing. Overall, the work reads more as a conceptual or idea paper than a genuine mechanistic discovery.

**Questions:**

like weakness

---

> ### Author Response · Authors · 2025-11-18
> **Response to Reviewer ei5w**
>
> We thank the reviewer for the feedback and would like to express our sincere gratitude for the time and effort you dedicated to reviewing our work. We would like to respond to your comments below:
>
>  - "This paper is conceptually interesting but offers limited substantive innovation. The proposed connection between transformer attention and the Encoding Specificity Principle is largely a loose analogy rather than a formal theoretical contribution.
>
> The analogy proposed in our paper, while conceptual, it is backed by empirical observations and experiments, as detailed in the paper. We employ rigorous experimental setups to observe behaviors in transformer models that align with the Encoding Specificity Principle. These empirical insights offer a foundational basis for the analogy, proposing that such parallels can be observed in practice.
> - "The authors do not provide a rigorous mathematical formulation or define concrete quantitative measures such as memory retrieval efficiency or cue–content overlap."
>
> While we didn't delve into creating new mathematical constructs, the empirical experiments of our paper offer quantitative data and statistical analyses that examine memory retrieval and cue-content dynamics under the proposed analogy. These analyses, we believe, lay a firm grounding for the conceptual claims made.
> - "As a result, the findings are primarily descriptive phenomena rather than statistically grounded or systematically analyzed results."
>
> Our empirical findings include well-defined experiments and statistical assessments designed to validate the conceptual framework. If these results were perceived as merely descriptive, we would appreciate specific feedback on sections that may require deeper quantitative focus or clarification to better illustrate the statistical grounding.
> - "Furthermore, the paper lacks stronger visualization or causal interpretability analysis. It does not present attention-head–level retrieval trajectories or activation dynamics that could substantiate the proposed analogy. Incorporating feature visualization or attention circuit tracing would make the conclusions considerably more convincing. Overall, the work reads more as a conceptual or idea paper than a genuine mechanistic discovery"
>
> The main findings in our paper involve the empirical results to validate our hypotheses. Simply put:
>   1. Q, K, and V play unique roles in attention layers that imitate the process of key-value memory.  Hence, memory should be impacted differently when different roles are impacted. The results of experiment 1 strongly support this separation of duties in QKV. Specifically, Figure 2 shows that when V is swapped, it introduces hallucination to the model, since V plays the role of content storage. On the other hand, swapping K, which indexes the content, does not introduce hallucination but impact memorization instead.
>   2. Contextual memory retrieval facilitated by QKV relies on keywords-as-cues to successfully retrieve. Our Figure 3 and Table 1 show the existent of 1 single layer-head-dimension (1 for each model) that is mainly responsible for retrieving these cues, which in turn enable extraction of what the models consider "cues". Figure 4 then shows the impact of omitting the attention others pay to these "cues" on memorization, with out method outperforming "cues" generated by GPT-4o and less harmful than a recent XAI method - LXT. Showing how these keywords indeed act as cues to enable a successful contextual memory retrieval.
>
> With the main objective of our paper being to validate our hypotheses, we thought the provided figures and causal interpretability analysis were sufficient. Agree that we did not "present attention-head–level retrieval trajectories or activation dynamics" but that is not the objective of the paper. Our work is motivated by machine unlearning but is not actively proposing an unlearning method, so we did not see the need to go down that path. If there are specific empirical sections you believe require more depth or visibility, please provide further guidance.

---

> ### Comment · Reviewer_ei5w · 2025-11-20
>
> The paper lacks evaluation on mainstream XAI datasets, like some standard ICL datasets, like [1], which I consider essential for a memory retrieval study. Therefore, I will maintain my original score.
>
> [1] Min, Sewon, et al. "Rethinking the role of demonstrations: What makes in-context learning work?." arXiv preprint arXiv:2202.12837 (2022).

---

> > ### Author Response · Authors · 2025-11-24
> > **Response to Reviewer ei5w**
> >
> > We thank the reviewer for the feedback.
> >
> > We would like to justify why we design our dataset that way and not utilize any other commonly used datasets in other XAI works. There are 2 points:
> >
> > $\textbf{1. Focus on Memory Rather Than General Explainability}$
> >
> > As discussed in Section 2.1, our work differs significantly from most other XAI approaches, which typically aim to explain how and why an AI system arrives at a particular output, often attributing the result to a combination of factors such as grammar, syntax, semantics, and other reasoning processes. In contrast, our study focuses exclusively on the memory aspect, isolating the extent to which the model’s output is influenced by its ability to recall specific information.
> >
> >    To this end, we select a dataset that best reflects a model’s memory: a text completion dataset in which each prompt elicits a verbatim response of sufficient length (in our work, it is 40 tokens) from the model. In other words, our dataset consists of prompts for which the model is able to reproduce the expected output word-for-word $\textit{(We described our filtering process to create this dataset in section 3.2)}$.
> >
> >    We believe this type of dataset best reflects a model's memory because it poses a particularly challenging task: the model must generate every single token exactly as in the expected output. In contrast, tasks such as question answering or classification often require only a few correct tokens, which may not provide as strong an indication of memorization. Moreover, producing the correct answer on a Q&A or classification task does not necessarily mean that the model’s memory was accurately accessed. While our approach cannot guarantee perfect measurement of memorization either, its higher difficulty gives us greater confidence that any correct output truly involves invoking the model’s memory.
> >
> > $\textbf{2. Selection of Book Content as Dataset}$
> >
> > Our selection of book content as the dataset was intentional. Unlike other text completion datasets, the content of books offers well-defined contextual boundaries that set it apart from sources such as news articles or encyclopedic entries (Wiki). The events, places, and characters found in books are typically unique to each work, making it easier to distinguish and analyze their context. This uniqueness allows us to more effectively compare the keywords we extract with those identified by other methods (see Table 1).

---

### Official Review · Reviewer_pQPV · 2025-10-30

**Soundness:** 3
**Presentation:** 3
**Contribution:** 2
**Rating:** 6
**Confidence:** 3

**Summary:**

This paper investigates memory mechanisms in Transformer-based Large Language Models (LLMs), with a specific focus on the role of attention layers. The authors propose a conceptual framework based on principles from human psychology, chiefly the Encoding Specificity Principle (ESP) and cue-based retrieval theories. This framework leads to two core hypotheses. The paper presents two main experiments to empirically validate these hypotheses using several decoder-only LLMs. The authors identify specific neurons that are highly activated by these keywords and show that perturbing the K-projection for these keywords significantly impairs memory recall far more than perturbing random tokens. The paper concludes that this evidence supports the ESP framework and identifies a pathway for extracting memory-indexing keywords, which could be used for applications like machine unlearning.

**Strengths:**

1. The paper's primary strength is its novel conceptual bridge, connecting the well-established psychological theory of cue-based retrieval and the Encoding Specificity Principle directly to the architectural components of the Transformer. This provides an intuitive and human-centric lens for interpreting the "black-box" attention mechanism.

2. This conceptual framework is supported by a very strict and rigorous experimental design. Experiment 1 is particularly well-controlled. It carefully isolates the roles of Q, K, and V by swapping them between factual and counterfactual prompts that are constrained to have the exact same tokenized length. The intervention is also minimal, applying only to the first token generation, which cleanly tests the effect of context processing.

3. Experiment 2 is equally rigorous. It validates the "keywords-as-cues" hypothesis not just by perturbing keywords, but by benchmarking this against a crucial control: perturbing an equivalent number of random tokens. The dramatic difference in outcomes, shown in Figure 4, strongly supports the claim that these keywords are functionally special.

**Weaknesses:**

1. A primary point of clarification is that the paper does not present a new formal, mathematical theory; rather, it provides an empirical validation of a conceptual mapping from psychology. Its support is based on experimental evidence, not mathematical proofs.

2. The authors themselves identify limitations in their methodology. For instance, the method for selecting top neurons and the number of keywords to target for perturbation is described as "naive" and "largely arbitrary," suggesting that the full potential of the unlearning application is not yet realized.

3. A further methodological limitation, also noted by the authors, is the inability of their keyword extraction method to handle compound words or multi-word terms as single cues. The paper notes that "White rabbit" is a better cue than "rabbit" alone, but the current method cannot group these tokens, meaning the extracted keyword list may not fully capture the ideal set of contextual cues. This could under-represent the true effect of these cues.

**Questions:**

1. The K-perturbation experiment successfully demonstrates that zeroing-out keywords impairs memory. How sensitive are these results to the type of perturbation? For instance, what would be the effect of replacing the keyword K-projections with random noise, or with an averaged K-vector from other keywords, instead of simply zeroing them out?

2. Figure 3 shows that for each model, a single layer-head-dimension triplet is consistently the most activated by keywords across different books. Does perturbing only this single, dominant neuron have a disproportionately large impact on memory recall? How much of the memory impairment effect is localized to this one neuron versus the other high-ranking neurons?

3. The paper provides strong evidence that the attention mechanism functions *like* a key-value memory. Does this framework offer any insights into how the model *learns* to associate specific keywords with the K-matrix (the index) during pre-training? Does this imply that the K-projections are specifically trained to act as a content-addressable index for salient tokens?

4. Finally, the experiments focus on factual recall from texts. How robust is the "keywords-as-cues" hypothesis to different types of memory? Would this same mechanism (keywords indexed by K) be expected to retrieve more abstract or thematic concepts, or is it specialized for concrete factual associations?

---

> ### Author Response · Authors · 2025-11-18
> **Response to Reviewer pQPV**
>
> We thank the reviewer for the feedback and would like to express our sincere gratitude for the time and effort you dedicated to reviewing our work. We would like to respond to your comments below:
>
> - "A primary point of clarification is that the paper does not present a new formal, mathematical theory; rather, it provides an empirical validation of a conceptual mapping from psychology. Its support is based on experimental evidence, not mathematical proofs"
>
>  Our paper is indeed empirical at large, there is no denying about that. But we also make our distinction with other XAIs and point out the current debate about attention's explanatory value depending on the goal of explain-ability, showing our view of explain-ability, which consists of many different linguistic aspects. Mathematical proofs from other XAIs might be able to provide interpretability value in overall, but not in one specific aspect that we focus on in this work - contextual memory. We hope that with this empirical study, future works that target memory aspect specifically can be enabled, knowing what roles Q, K, and V play in it, and what facilitate the retrieval process.
> - "The authors themselves identify limitations in their methodology. For instance, the method for selecting top neurons and the number of keywords to target for perturbation is described as "naive" and "largely arbitrary," suggesting that the full potential of the unlearning application is not yet realized."
>
> Even though our work is motivated by machine unlearning, it is not our objective in this work. We also argue how the lack of understanding of transformer's memory mechanism is hindering the development of machine unlearning field and we hope to address that in this work.
> - "A further methodological limitation, also noted by the authors, is the inability of their keyword extraction method to handle compound words or multi-word terms as single cues. The paper notes that "White rabbit" is a better cue than "rabbit" alone, but the current method cannot group these tokens, meaning the extracted keyword list may not fully capture the ideal set of contextual cues. This could under-represent the true effect of these cues."
>
> While the limitation is true, it mainly concerns the experiments we perform in this work. This is due to the fact that human use "word" as a single linguistic unit, and LLMs (majority) use "sub-word token". Therefore, to compare our extracted keywords with ones generated by GPT-4o, we need to combine these sub-word tokens to word units for a fair comparison and for us to evaluate the semantical meanings.
>
> Now that we successfully identified the layer-head-dim triplets that correspond to these keywords, we can use them to extract sub-word tokens for unlearning purpose without having to convert to words.
> - "The K-perturbation experiment successfully demonstrates that zeroing-out keywords impairs memory. How sensitive are these results to the type of perturbation? For instance, what would be the effect of replacing the keyword K-projections with random noise, or with an averaged K-vector from other keywords, instead of simply zeroing them out?"
>
> The choice of zeroing-out perturbation is to reflect the reality where other words do not pay attention to keywords. We did not perturb them in any other ways because there is no intuitive reasoning we can think of that would make sense in terms of memory. For example, with random noise, other words now pay random attention to keywords, what would that mean ? and how can we even interpret the results ?. Not to mention that our perturbation happens before softmax calculation, so if the attentions paid to keywords by other words are large enough, we envision that the retrieval process will still happen with minimal impairment.
>
> Agree that this is a very interesting aspect that you point out, which we will think about in future work.
>
> (Continue in next comment)

---

> ### Author Response · Authors · 2025-11-18
> **Response to Reviewer pQPV (continue)**
>
> - "Figure 3 shows that for each model, a single layer-head-dimension triplet is consistently the most activated by keywords across different books. Does perturbing only this single, dominant neuron have a disproportionately large impact on memory recall? How much of the memory impairment effect is localized to this one neuron versus the other high-ranking neurons?"
>
> We thought of the same when observing the results. We then perform perturbation of a fix proportion of top layer-head-dimension triplets corresponding to the keywords, but the impairment effect is negligible. So this means that if you prune the top triplet, the second best one will simply take it place. This is accordance with the linear superposition hypothesis where multiple features overlap within same neurons [1][2][3].
> - "The paper provides strong evidence that the attention mechanism functions _like_ a key-value memory. Does this framework offer any insights into how the model _learns_ to associate specific keywords with the K-matrix (the index) during pre-training? Does this imply that the K-projections are specifically trained to act as a content-addressable index for salient tokens?"
>
>  It does indeed provide insights into how model learns during pre-training. In the Encoding Specificity Principle's words mentioned in our paper - "successful retrieval depends on the overlap between contextual cues present at encoding and those available at retrieval". Encoding here is referring to the training process of model, where it learns/stores memory by encoding these keywords. Then during retrieval, model looks for these keywords to recall memory.
>
> Regarding your second question, our experiment results indeed imply this fact. Not only that but V-projections are also implied to specifically trained to act as the storage of the content.
>
> Furthermore, with the discovery of a model (Olmo-2) that strongly relies on keywords to recall memory, future work can further experiment on this model specifically to understand why it is different from others. This can potentially show what exactly happen during the training for the model to behave that way.
> - "Finally, the experiments focus on factual recall from texts. How robust is the "keywords-as-cues" hypothesis to different types of memory? Would this same mechanism (keywords indexed by K) be expected to retrieve more abstract or thematic concepts, or is it specialized for concrete factual associations?"
>
> In our experiments, we show that keywords can serve as effective retrieval cues for information that is explicitly linked to them in the text, which is likely more robust for factual recall. On the other hand, thematic or conceptual concepts go beyond word-fact associations, so it is less likely our "keywords-as-cues" works for this type of memory. Again referring back to our sole target of contextual memory in this work for machine unlearning purpose.
>
> [1]Sanjeev Arora, Yuanzhi Li, Yingyu Liang, Tengyu Ma, and Andrej Risteski. 2018. Linear algebraic structure of word senses, with applications to polysemy. Transactions of the Association for Computational Linguistics, 6:483–495
> [2]Chris Olah, Nick Cammarata, Ludwig Schubert, Gabriel Goh, Michael Petrov, and Shan Carter. 2020. Zoom in: An introduction to circuits.
> [3]Nelson Elhage, Tristan Hume, Catherine Olsson, Nicholas Schiefer, Tom Henighan, Shauna Kravec, Zac Hatfield-Dodds, Robert Lasenby, Dawn Drain, Carol Chen, Roger Grosse, Sam McCandlish, Jared Kaplan, Dario Amodei, Martin Wattenberg, and Chris Olah. 2022. Toy models of superposition. Transformer Circuits.

---

### Official Review · Reviewer_Ly4f · 2025-10-31

**Soundness:** 1
**Presentation:** 2
**Contribution:** 1
**Rating:** 2
**Confidence:** 3

**Summary:**

This paper studies the hypothesis that the attention mechanism in Transformer implements memory-like functions analogous to those found in human cognition. The evidence for this hypothesis comes from two experiments. First, they swap the attention activations between two counterfactual prompts and observe the resulting outputs. Second, they decode and compare lists of keywords from the attention heads for different documents. Experimental results suggest that the attention mechanism performs and information retrieval role.

**Strengths:**

- Establishing similarities between artificial intelligence and biological intelligence is an important and interesting direction.
- The analogy between transformers and cue-based retrieval is clearly explained.
- The authors state their hypothesis clearly and support them with experiments.

**Weaknesses:**

- The paper appears to be concealing the important distinction between long-term and short-term memory. This makes the argued similarity appear somewhat odd. LLM short-term memory (prompt) si compared with human long-term memory (hippocampal subregion).
- An important motivation of the paper is *machine unlearning*, but this is usually a concern with regard to the LLM long-term memory (weights), not the short-term memory (prompt) studied in this paper.
- The experimental results for attention swapping are not surprising, mirroring many of the already existing causal interventions in the XAI literature.
- Understanding transformers from a memory-retrieval perspective is not a novel idea. [1]

[1] Bietti, Alberto, et al. "Birth of a transformer: A memory viewpoint." Advances in Neural Information Processing Systems 36 (2023): 1560-1588.

**Questions:**

- I see that the Encoding Specificity Principle is concerned with episodic (long-term) memory. Can the authors point to any evidence for a retrieval-like mechanism in human **short-term** memory?
- Do the authors see a way that their methods (or similar) could be applied for unlearning?

---

> ### Author Response · Authors · 2025-11-18
> **Response to Reviewer Ly4f**
>
> We thank the reviewer for the feedback and would like to express our sincere gratitude for the time and effort you dedicated to reviewing our work.
> We would like to respond to your comments below:
>
> - "The paper appears to be concealing the important distinction between long-term and short-term memory. This makes the argued similarity appear somewhat odd. LLM short-term memory (prompt) si compared with human long-term memory (hippocampal subregion)." +
>   "An important motivation of the paper is _machine unlearning_, but this is usually a concern with regard to the LLM long-term memory (weights), not the short-term memory (prompt) studied in this paper."
>
>  Since the very beginning, we draw an important motivation to our work for machine unlearning, so we are solely focusing on long-term memory of LLM, not short-term memory in prompt of in-context learning.
>   Our experiments are also designed to probe this long-term memory directly.
>
>   For example, attention swapping experiment (Section 4.1 and Figure 2) swap weight matrices projections between prompts.
>   K matrix weight perturbation experiment (Section 4.2 and Figure 4), even though it does not directly change weights, the perturbation process prevent projections of keywords (by setting projection values to 0), which in turn facilitate the reality where other words in the prompt do not pay attention to the keywords.
> - "The experimental results for attention swapping are not surprising, mirroring many of the already existing causal interventions in the XAI literature."
>
>  In Section 2.1, we provide a background of existing XAI literatures and how our work is very different from the rest. We also mentioned in the same section how attention layers are often neglected with debate about their explanatory value.
>
> Motivated by machine unlearning objective, we argued that "Beyond much of the conventional XAI literature, which seeks comprehensive explanations involving grammar, syntax, and semantics, our work specifically targets memory mechanisms in transformer models, emphasizing memory as one distinct and influential component that contributes to the accurate and human-like responses from LLMs."
>   To the best of our knowledge, we are the first to view explain-ability in AI this way and provide empirical evidences for our 2 hypotheses that target the memory mechanism specifically. If there exist such XAI works that share the same view which we overlook, we would really appreciate some pointers.
> - "Understanding transformers from a memory-retrieval perspective is not a novel idea"
>
>  As we discussed in our Section 2 (and more specifically Section 2.2-2.3), attention mechanism in transformers has long been theorized to function like memory-retrieval process, especially with the dot-product calculation being a similarity measurement.
>
> But to what extend each component Q, K, and V plays in this process if under-explored. Most works provide evidences to their view by employing technique like toy model or in the paper you referenced - a simplified two-layer transformer. While we directly use pre-trained LLMs in the wild, which provide a much more realistic view.
>
> Not to mention the our target for memory is very unique where we aim to understand how memory of a context is stored (contextual memory). In the paper you referenced, the authors are more concerned with "associative memory" that is often found in in-context learning.

---

> ### Comment · Reviewer_Ly4f · 2025-11-18
>
> I thank the authors for their time and effort of replying to my critiques. After reading the response, I became convinced that authors misunderstand the distinction between long- and short- term memory in LLMs. In consequence, I have updated my score to a strong reject. I will explain my reasoning in detail below in hopes that authors will understand my position and will not become discouraged.
>
> # Attention as Short-Term Memory Retrieval
>
> Regarding this statement:
>
> > we are solely focusing on long-term memory of LLM, not short-term memory in prompt of in-context learning
>
> Let me clarify what I mean by long and short term memory in LLMs. By short-term, I mean the prompt, and any factual information that is present in the prompt. By long-term memory, I mean any factual information from the training data that has become embedded in the weight matrices during training. I hope the authors agree with these standard, common-sense definitions.
>
> Your hypothesis reads:
>
> > **Hypothesis 1.** Q encodes retrieval cues, K indexes candidate traces by those cues, and V stores
> retrievable content.
>
> Where I assume Q, K, V are the queries, key, and values in self-attention. Note, however, that the keys, values, and queries are computed *entirely* based on the prompt in order for certain tokens to attend to other tokens in the prompt. The self-attention mechanism is *only useful* to retrieve factual information from the same prompt. The self-attention cannot retrieve information from the training data. Hence, your hypothesis is concerned entirely with short-term memory.
>
>  # Novelty and Existing literature
>
> > In Section 2.1, we provide a background of existing XAI literatures and how our work is very different from the rest. We also mentioned in the same section how attention layers are often neglected with debate about their explanatory value.
>
> This sentence from Section 2.1 has drawn my attention as a novelty claim:
>
> > However, existing work has mainly focused on Feed-Forward Neural Network (FFNN) layers.
>
> Hence, the authors claim to provide a novel perspective into long-term memory/unlearning based on the self-attention layer, as opposed to existing work focused on FNNNs (aka Multi Layer Perceptrons -- MLPs). However, there is a fundamental problem here. The MLPs are the **only** module capable of memorizing raw factual information from the training data, hence long-term memory. This is because, as I argue in the previous section, self-attention is mostly useful for in-context retrieval.
>
> When it comes to the self-attention mechanism, there has been extensive interpretability work [1] using the method commonly known as *activation-patching* for attention heads, which is very similar to what the authors do in their first experiment. I say the first experiment is not too surprising because patching all keys and values is essentially the same as placing the token in the counterfactual prompt.
>
> Hence, the experiment is not novel, and it is in support of a very problematic hypothesis.
>
> # Closing Remarks
>
> I truly hope that authors do not become discouraged by my negative review. I wish they continue to study interpretability of AI models, and in particular the connections between interpretability and the psychological literature.
>
> [1] Li, Maximilian, and Lucas Janson. "Optimal ablation for interpretability." Advances in Neural Information Processing Systems 37 (2024): 109233-109282.

---

> ### Author Response · Authors · 2025-11-18
> **Response to Reviewer Ly4f**
>
> We thank the reviewer for the swift response and the detailed comments.
>
> We would like to address your concern as followed:
>
> - "By short-term, I mean the prompt, and any factual information that is present in the prompt. By long-term memory, I mean any factual information from the training data that has become embedded in the weight matrices during training"
>
> Yes, I absolutely agree with you on these definitions, perhaps our use of the terms $Q$, $K$ and $V$ appear to be a bit confusing (we follow the same notion as the "Attention is all you need" paper, where they use $QKV$). When we say $Q$, $K$ and $V$ in our Hypothesis 1, we mean the attention weight matrices learned during training at attention layers, aka the weights that transform the initial embeddings to $Q$, $K$, and $V$ (again, following the notion in "Attention is all you need"), which are later used in the self-attention calculation. We will update our paper to use clearer notation to avoid this confusion.
>
> We specifically call our study's target *Contextual memory* because we agree with you that perhaps the actual memory is being recalled in FFNN layers (as suggested by [1][2]). But due to the linear and residual information flow nature of transformers, the memory recall process in FFNN is highly influenced by attention layers. We theorized that the attention layers provide contextual memory or in other words, help model understand the meaning of words, in the form of attention. And only after such contextual memory is supplied, does the memory retrieval process in FFNN become successful.
>
> We want to further highlight that *contextual memory* that we are interested in, is absolutely a form of long-term memory, not short-term. Going back to the calculation of attention:
>
> $Attention = softmax(\frac{QK^T}{\sqrt{d_k}})V$
>
> Where $Q=EW^Q$, $K=EW^K$, and $V=EW^V$, with $E$ being the input embedding from input $x$, and $W$ are the learned weight matrices in attention layers. The projected values from embedding $E$ to $Q, K$ and $V$ are long-term memory. The dot-product $\frac{QK^T}{\sqrt{d_k}}$ is short-term.
>
> Our experiment 2 does exactly that, where we choose to set the projected values of keywords by $K$ weight matrix to 0, effectively setting the projection of input embedding with $K$ weight matrix to 0 where keywords are. Which in turn, when calculating self-attention (dot product with $Q$), making the calculation as if other words pay NO attention to the keywords, theoretically giving those keywords no meaning.
>
> Regarding your concern related to our experiment 1, we agree with the similarity you pointed out between our experiment 1 and activation-patching, but even within ablation study using activation-patching, different patching locations yield different interpretability results. For our experiment, we isolate this effect to individual attention weights to study their roles in the process of *contextual memory* retrieval.
>
> The swapping experiment targets the projected values $Q$, $K$, and $V$, which are long-term memories between prompts in a pair of factual-counterfactual prompts.
>
> So our experiment to "patching all keys and values" is not the same as placing the token in the counterfactual prompt, because the queries (or $Q$) remain unchanged. Not to mention that we also conduct swapping of other projected values too that empirically show different interpretability (For example, swapping $V$ means $Q$ and $K$ remain unchanged, yet still induce hallucination highly in model), and our experiment 2's results.
>
> We hope that this clear our misunderstanding.
>
> [1]Mor Geva, Roei Schuster, Jonathan Berant, and Omer Levy. Transformer feed-forward layers
> are key-value memories. In Marie-Francine Moens, Xuanjing Huang, Lucia Specia, and Scott
> Wen-tau Yih (eds.), Proceedings of the 2021 Conference on Empirical Methods in Natural Language Processing, pp. 5484–5495, Online and Punta Cana, Dominican Republic, November
> 2021. Association for Computational Linguistics. doi: 10.18653/v1/2021.emnlp-main.446. URL
> https://aclanthology.org/2021.emnlp-main.446/.
>
> [2]Mor Geva, Avi Caciularu, Kevin Wang, and Yoav Goldberg. Transformer feed-forward layers
> build predictions by promoting concepts in the vocabulary space. In Yoav Goldberg, Zornitsa
> Kozareva, and Yue Zhang (eds.), Proceedings of the 2022 Conference on Empirical Methods
> in Natural Language Processing, pp. 30–45, Abu Dhabi, United Arab Emirates, December
> 2022. Association for Computational Linguistics. doi: 10.18653/v1/2022.emnlp-main.3. URL
> https://aclanthology.org/2022.emnlp-main.3/.

---

### Official Review · Reviewer_DMUK · 2025-11-05

**Soundness:** 2
**Presentation:** 2
**Contribution:** 2
**Rating:** 2
**Confidence:** 3

**Summary:**

The paper presents two hypothesis and experiments around them.

H1 : Q is context encoder, K as trace memory index, and V as content store (This is obvious in how the Q,K,V are named)

H2 : Encoding Specificity Principle: Basically most effective circumstances for retrieval is most prominent cues during encoding are available at retrieval, so they believe this would be keywords

The findings were not new and re-iterate what's already known.

**Strengths:**

Encoding-Specificity Argument, from a conceptual standpoint the arguments and experiments they conduct are sound and logical, rigorous testing using 6 different models.
The figures are clear in displaying the information and takeaways. Generally good contextualization before each figure as well.

**Weaknesses:**

The first experiment has very significant perturbation so the significance of their results does not support their idea too much. For the first experiment, they do not talk about how they swap the Q, K, and V matrices. So if the matrix V is swapped, what is it swapped with?

Why does is H1 even a hypothesis ? (H1 - Q encodes retrieval cues, K indexes candidate traces by those cues, and V stores
retrievable content.) Isn't this why the matrices are called Query, Key and Value matrices?

For H2 : The method for finding keywords is unclear. Also putting key-vector activations of certain keywords is same as zeroing out their attention scores.

Overall, the experiments are not explained in enough detail. For H1 experiments, it would be nice to see examples of counterfactual and fact pairs used. The biggest weakness would be that unfortunately we don't learn anything new from the results of the experiments.

**Questions:**

See weaknesses

---

> ### Author Response · Authors · 2025-11-18
> **Response to Reviewer DMUK**
>
> We thank the reviewer for the feedback and would like to express our sincere gratitude for the time and effort you dedicated to reviewing our work.
> We would like to respond to your comments below:
>
> - "For the first experiment, they do not talk about how they swap the Q, K, and V matrices. So if the matrix V is swapped, what is it swapped with?"
>
> In Section 3.1, the paragraph right after the description of Dataset is where we describe how swapping is conducted.
>   An example of this swapping might clear the confusion. Let's say we have a pair of prompts (with their respective answers) from the Counterfact dataset:
>
>   Prompt A: "What is the capital of Japan" - Answer A: "Tokyo"
>   --> The model creates $Q_a-K_a-V_a$
>
>   Prompt B: "What is the capital of France" - Answer B: "Paris"
>   --> The model creates $Q_b-K_b-V_b$
>
>   Either prompt in this case can act as factual and the other counterfactual (But let say A is factual and B is counterfactual).  Then, during swapping, with swapping target being $V$, prompt A after being input will have its $V_a$ swapped for $V_b$.
>
>   --> The calculation of attention for input prompt A becomes: $Q_a-K_a-V_b$ instead of $Q_a-K_a-V_a$
>
>   Hope that this also address the reviewer's concern about examples of counterfactual and fact pairs used.
> - "The first experiment has very significant perturbation so the significance of their results does not support their idea too much"
>
> In the same paragraph mentioned above, we also describe how our swapping "only occurs for the input prompts, not the subsequent generated text". Essentially, using the cue-based retrieval analogy to describe this process, we only slightly point the retrieval process at the beginning to a new memory slot.
> - "Why does is H1 even a hypothesis ? (H1 - Q encodes retrieval cues, K indexes candidate traces by those cues, and V stores retrievable content.) Isn't this why the matrices are called Query, Key and Value matrices?"
>
>  Attention mechanism in transformers proposed in the "Attention is all you need paper" was indeed designed to reflect this process where they say "An attention function can be described as mapping a query and a set of key-value pairs to an output, where the query, keys, values, and output are all vectors". Note how the quoted sentence is the only part where they talked about the motivation behind the design, no mathematical evidences, no past works motivation. But design is only one part, LLMs are trained to update their weights  to potentially imitate this process. We do not know if weights learnt during training actually reflect this process, hence the reason why we went through great length pointing out the fact that LLMs are black-box.
>   We included past papers that share the same idea but they only manage to validate such process using toy model [1], not pre-trained LLM.
>   Furthermore, our experiment 1 also shows to what extend do each component of attention (Q, K, V) facilitate this process with the results of swapping only V and K-V together. As an example, we now have evidence that V stores content (or memory), so a task of unlearning should not involve V. But for a task of model editing (edit knowledge of model), then V should be the primary target.
> - "For H2 : The method for finding keywords is unclear. Also putting key-vector activations of certain keywords is same as zeroing out their attention scores."
>
>  We describe this process of finding keywords and finding layer-head-dimension triplet that correspond to these keywords in the paragraphs after Dataset in Section 3.2
>
>  Perhaps we should make it clearer in both Section 3.1 and 3.2 about our experiments description, potentially putting them in their own separate sections. Further feedbacks on how we should present these sections are greatly appreciated.
>
>
> [1]Samuel J. Gershman, Ila Fiete, and Kazuki Irie. Key-value memory in the brain. Neuron, 113(11):1694–1707.e1, 2025. ISSN 0896-6273. doi: https://doi.org/10.1016/j.neuron. 2025.02.029. URL https://www.sciencedirect.com/science/article/pii/ S0896627325001722.

---

### Meta-Review · Area_Chair_6Fo4 · 2026-01-13

**Summary:**

The authors study whether attention mechanisms implement memory-like functions similar to those found in human cognition. They present two hypotheses: 1) Q encodes retrieval cues, K indexes candidate traces by those cues, and V stores
retrievable content; 2) The retrieval cues are instantiated as salient lexical tokens (“keywords”) tied to the relevant memory. Reviewers expressed skepticism about the scope of the contributions and novelty relative to prior work.

**Reviewer Concerns:**

Reviewers generally seemed unconvinced by the responses, and were negative overall. It is unlikely that discussions would have been fruitful.

**Reviewer Scores:**

No significant changes.

---

### Decision · Program_Chairs · 2026-01-26

Reject